# Optimization and Preparation of Tallow with a Strong Aroma by Mild Oxidation

**DOI:** 10.3390/molecules27249047

**Published:** 2022-12-19

**Authors:** Yanjing Jin, Junaid Raza, Huanlu Song, Lijin Wang, Qiaojun Wang, Guoli Ma, Yang Xiao

**Affiliations:** 1Laboratory of Molecular Sensory Science, School of Food and Health, Beijing Technology and Business University (BTBU), Beijing 100048, China; 2Guanghan Maidele Food Co., Ltd., Guanghan 618305, China; 3School of Light Industry, Beijing Technology and Business University (BTBU), Beijing 100048, China

**Keywords:** tallow with a strong aroma, mild oxidation, aroma compounds, gas chromatography-olfaction–mass spectrometry (GC-O–MS)

## Abstract

This study was performed to extract and separate the volatiles with solid-phase microextraction (SPME), and was conducted to analyze volatile odor compounds qualitatively and quantitatively in the production of a strong aroma tallow by mild oxidation. A total of 51 odor compounds were detected in the tallow smelted under different conditions. It was found that the high proportion of aldehydes was an important feature of the aroma components in the oxidized melted tallow, such as 1-hexanal, heptanal, nonanal, octanal, benzaldehyde, etc. Through the determination of various indicators, sensory evaluation, and the gas chromatography-olfaction–mass spectrometry (GC-O–MS) analysis and, in combination with response surface methodology, the optimal process parameters for oxidative smelting of tallow were determined as follows: a reaction temperature of 149.61 °C, a reaction time of 31.68 min, and an airflow rate of 97.44 L/h. The accelerated oxidation test further verified the quality of the oxidized tallow.

## 1. Introduction

Tallow is a form of rendered fat and is classically made from beef. When the slaughtered cattle are raised to the slaughter condition, a large amount of adipose tissue will be accumulated under the epidermis, between muscles, in the abdominal cavity, intestinal region, and other parts, and the tallow adipose tissue obtained by segmentation is refined by using an appropriate processing method to obtain edible grease, which is called tallow [1].

Tallow is widely used in food processing because of its good flavor and is often used as the processing raw material of shortening, cakes, and other foods [2]. As the soul of the Sichuan and Chongqing hotpot, tallow played an important role in the formation of the hotpot flavor, both in its own flavor and the flavor changes produced during the cooking process [3]. According to the previous analysis of the aroma of 41 old hot pot tallow, it was found that the strong aroma tallow was particularly popular. Therefore, in this experiment, the strong aroma tallow was prepared through the mild oxidation of fat to meet the needs of consumers [4].

Lipid oxidation is an important pathway for the formation of characteristic meat flavor compounds. The research by Pearson et al. [5] has shown that the basic meat flavors of all kinds of meat flavors are the same, while the differences in characteristic meat flavors mainly come from the thermal oxidation degradation reaction of fat, in which heating-induced oxidation of unsaturated hydrocarbon chains in lipids is an important pathway and generates a variety of volatile compounds, including fatty aldehydes, ketones, and acids. Oxidation of fat at a normal temperature will produce a rancid taste, while oxidation under heating conditions will produce characteristic flavor substances, which are the main basis for adding fat and regulating fat oxidation in the production process of meat flavor essence. Therefore, in order to obtain the oxidized fat rich in volatile compound components, air heating can be used to regulate fat oxidation. The thermal reaction meat flavor product prepared by adding oxidized fat has the advantages of strong aroma, prominent characteristic flavor, etc. [6]. Due to the complexity of lipid oxidation and the uncertainty of factors, as well as the limitation of analytical methods, the research progress on the regulation of the preparation of flavor precursors from oxidized fat is relatively slow, but some research results have been achieved.

The research on the regulation of fat on oxidation is not systematic and it has only been reported in the early patents. For example, Aishima and Nobuhara [7] treated tallow with air heating under conditions of 150–170 °C, and then mixed and heated the obtained oxidized tallow with fermented soy sauce to prepare the roast beef essence. Haring [8] controlled the triglyceride oxidation process by altering the moisture and metal ions in the system and prepared the flavor concentrate by mild oxidation. Haring [9] prepared milk flavor by mildly oxidizing milk fat to increase the content of 3-methylbutanal and 2-nonenal. There were many studies on the regulated oxidation of fats in China. In general, there were mainly two oxidation methods: high-temperature heating-controlled oxidation and enzymatic mild oxidation. There were differences in the fragrance of the products obtained from the thermal reaction of these two kinds of oxidized fats, with the former being the “barbecue” flavor, and the latter being the “stewed” flavor [10]. For example, Li [11] oxidized tallow at 160–170 °C for 40 min and then thermally reacted with fermented soy sauce and cysteine to produce beef essence. Xiao and Sun [12] explored the regulation of tallow oxidation by response surface methodology and determined the optimal process conditions as follows: a reaction temperature of 158.2 °C, a reaction time of 5.28 h, an airflow rate of 1.49 L/min, and an antioxidant V_E_ addition of 0.15%. Ouyang et al. [13] subjected the fat to an air oxidation treatment to obtain a peroxide value of 8–10 meq/kg and then added the peroxide value into a thermal reaction system to prepare the fat essence. Sun et al. [14] studied the regulation of tallow oxidation through single factor experiment and orthogonal test and concluded that the optimal process conditions were as follows: a reaction time of 3 h at 140 °C, an airflow rate of 0.018–0.035 m^3^/h 100 g fat, and an added antioxidant V_E_ of 0.01%. Yang [15] took vegetable oil (sunflower oil) and animal oil (chicken fat) as raw materials, and prepared the Qing Xiang flavoring agent and meat precursor after catalytic oxidation with lipoxygenase. Zhong et al. [16] used lipoxygenase to catalyze the oxidation of chicken fat. By measuring the peroxidation value of chicken fat and examining the effects of time, substrate (chicken fat) concentration, buffer pH value, and concentration on the oxidation of chicken fat, they finally determined the optimal oxidation process parameters. Yan and Tang [17] treated chicken fat with Novozyme lipase and added it into the Maillard reaction system to prepare chicken essence.

At present, research on the process of beef tallow flavoring is still relatively scarce. The butter used in the Sichuan and Chongqing hotpot is mainly refined butter. After deacidification, decoloration, deodorization, refining, and other production processes, the crude butter has a clear color and mellow flavor. It contains a variety of volatile flavor components, such as aldehydes, alcohols, ketones, and other compounds. These substances continue to react with protein and amino acids in tallow to produce a certain special odor, and form the unique volatile flavor of tallow [18]. The unique flavor of tallow mainly comes from aldehydes, alcohols, esters, ketones, and so on, and the formation process of flavor substances is complicated and is influenced by many conditions. In order to produce strong-flavor beef tallow, a combination of mild oxidation technology was used to obtain a beef tallow product with a stronger aroma than common beef tallow so as to optimize its flavor. Some lipid oxidative degradation and a Maillard reaction will occur in the oxygen evolution process, which will change the type and content of volatile compounds to a certain extent and cause flavor changes [19].

In view of the above background, solid-phase microextraction (SPME) was mainly used to extract and separate the volatiles in this study and then the gas chromatography-olfaction–mass spectrometry (GC-O–MS) technology was used to qualitatively and quantitatively analyze the aroma active compounds. The changes in aroma compounds in the oxidation of beef tallow were studied. Combined with sensory evaluation, the optimal conditions were determined, which provided a scientific basis for exploring the development and utilization of Nongxiang tallow hot pot seasoning, quality control, and use safety.

## 2. Results and Discussion

### 2.1. Effects of Different Factors on AV, PV, and P-AV of Oxidized Melted Tallow

When the temperature is 150 °C, the time is 35 min, and the airflow rate is 108 L/h (Figure 1, Figure 2 and Figure 3), the hydroperoxide, the primary oxidation product of tallow, reaches the maximum value. When it exceeds this value, the tallow oxidation is already in the termination stage. It can be seen in the figure that the PV and P-AV values change significantly beyond a specific node, so it is speculated that the oxidation induction period should be less than 35 min and the reaction is already in the oxidation transfer period; that is, the generation of hydroperoxide in the primary oxidation reaction and the pyrolysis of hydroperoxide in the secondary oxidation reaction are both increasing significantly, but the primary oxidation reaction is the main reaction, so both PV and p-AV values increase. At 150 °C, the primary oxidation product of tallow hydroperoxide reached the maximum value. When the temperature exceeds 150 °C, the oxidation of tallow was in the end stage and the generation of secondary oxidation products, namely the pyrolysis reaction of hydroperoxide, became the leading reaction and the generation of primary oxidation products took a back seat, so the PV value began to decline while the P-AV value increased significantly. From the aspect of the reaction mechanism, the higher the melting temperature and oxygen supply, the lower the AV value. It reached the lowest point at 150 °C and 108 L/h, which may be due to the oxidation of unsaturated fatty acid to generate hydroperoxide. Considering the influence of oxidation temperature on the PV, P-AV, and AV value, the oxidation melting of tallow at 150 °C for 35 min and an airflow rate of 108 L/h was selected.

### 2.2. Sensory Evaluation of the Sample

The weighted scores of each flavor type of melted tallow were different under different conditions (Table 1). Under reaction temperature conditions, the weighted total score was a maximum of 2.54 at 150 °C and a minimum of 2.14 at 120 °C. The milk flavor, sweet taste, and fruit flavor were the most intense at 150 °C, while each flavor type was relatively weak at 120 °C. In addition, the animal fat at 140 °C and 160 °C had a thick taste but, at the same time, the acid odor also increased. Under the reaction time conditions, the weighted total score for 35 min was the highest at 3.29, and the weighted score for 65 min was the lowest at 1.83. Milk flavor, animal fat flavor, and sweet taste were the most prominent at 25 min and 35 min, while each flavor type was relatively weak at 65 min. Under the airflow rate condition, the weighted total score of 108 L/h was the highest at 3.12 and the weighted total score of 138 L/h was the lowest at 1.83, of which the “shan” flavor and sweet taste at 108 L/h were higher than those under other flow rate conditions, while each flavor type at 138 L/h was relatively weak.

### 2.3. SPME-GC–MS Analysis Results of Oxidized Melted Tallow under Different Conditions

SPME-GC–MS was used for qualitative and quantitative analysis of compounds with aroma characteristics. A total of 51 aroma compounds were detected in tallow melted under different conditions (Table 2, Table 3 and Table 4). Among them, twenty-nine aroma compounds, including fifteen aldehydes, four alcohols, four acids, three esters, and three heterocyclic compounds were detected in tallow melting at different reaction temperatures. A total of thirty-nine aroma compounds were detected in melted tallow at different reaction times, including seventeen aldehydes, six acids, six ketones, five esters, two alcohols, and three heterocyclic compounds. A total of twenty-nine aroma compounds, including fourteen aldehydes, eight acids, three alcohols, three esters, and one other compound were detected in tallow melting at different air flow rates.
(1)Qualitative results

The aromatic compounds in the melted tallow were different under different conditions. The compounds 1-pentanal (almond), 1-hexanal (green, grassy), 1-heptanal (fatty), 1-octanal (fatty), *(E)*-2-heptenal (fatty), *(E)*-2-decenal (fatty), 1-nonanal (fresh), *(E)*-2-nonenal (cucumber), *(E)*-2-octenal (cucumber), *(E,E)*-2,4-decadienal (fried), 1-pentanol (spicy), 1-octen-3-ol (mushroom), benzaldehyde (nutty), 2-undecenal (orange peel), acetic acid (sour), octanoic acid (rot), and decanoic acid (putrid) are all contained in the melted tallow under different conditions, which are the basic components of the fragrance of oxidized melted tallow. Differences in other components led to differences in the aroma of oxidized melted tallow under different conditions.
(2)Quantitative results

A heat map was drawn according to the content of aroma compounds in the oxidized melted tallow under different conditions, as shown in Figure 4. In the figure, blue represents content below the average level, red represents content above the average level, and red also indicates higher content.

It could be seen that the overall content of aldehydes at 120 °C was higher than other temperatures in the reaction temperature, especially 1-hexanal (green, grassy), 1-nonanal (fatty), 1-octanal (fatty), *(E)*-2-nonenal (cucumber), *(E)*-2-decenal (fatty) and 2-undecenal (orange peel). The 1-heptanal (fatty), benzaldehyde (nutty), and 2-methylpyrazine (nutty) at 150 °C were more prominent. However, according to the sensory evaluation and determination of the oxidation index, it was found that although the overall substance content was high at 120 °C, the flavor was not very popular, largely because the temperature did not reach the melting point during the melting process, resulting in a slight fishy and greasy taste of the overall flavor and, thus, the aldehyde content was higher than that of samples at other temperatures.

The overall content of aldehydes was the highest in the reaction time of 65 min, especially 1-nonanal (fresh), *(E)*-2-nonenal (cucumber), *(E)*-2-decenal (fatty), 2-undecenal (orange peel), and *(E,E)*-2,4-decadienal (fried). In addition, it also contains high acid substances, such as acetic acid (vinegar), octanoic acid (putrid), nonanoic acid (waxy), etc. At the same time, it added many ketone substances, such as 2-octanone (earthy), 2-nonone (refreshing), 2-decanone (orange peel), etc. However, according to the sensory evaluation and determination of the oxidation index, it was found that although the overall substance content of beef tallow under 65 min was higher than that of samples under other conditions, the long reaction time further oxidized and degraded the fat and produced more small molecule aldehydes, ketones, and acid compounds, thus showing serious sour and rancid odor.

The total substance content of 108 L/h in the airflow rate was relatively high, especially 1-hexanal (green, grassy), 1-heptanal (fatty), 1-octanal (fatty), 1-nonanal (fresh), *(E)*-2-decenal (fatty), *(E,E)*-2,4-decadienal (fried), 2-undecenal (orange peel), α-pentyl cinnamic aldehyde (floral), etc. According to the sensory evaluation and determination of the oxidation index, it was found that the overall substance content of the 108 L/h melted tallow was not too high and it had the characteristic flavor of tallow, such as fat aroma.

### 2.4. Composition Characteristics of Aroma Compounds in Oxidized Melted Tallow under Different Conditions

By calculating the proportion of various aroma compounds in the melted tallow under different conditions (Table 5), it was found that the high proportion of aldehydes was an important feature of the aroma components in the oxidized melted tallow. The reaction temperature was kept within the range of 140–160 °C, the reaction time was within the range of 25–45 min, and the airflow rate was within the range of 78–138 L/h. The proportion of aldehydes was the highest, with an average of 53.46%.

### 2.5. Principal Component Analysis

In order to further clarify the aroma characteristics of oxidized melted tallow under different conditions, principal component analysis was performed on 15 levels of oxidized melted tallow with different factors according to the compositions and contents of different aroma compounds, as shown in Figure 5. It could be seen from the figure that the samples of 15 levels of oxidation-smelting tallow with different factors were relatively concentrated and could be divided into three types after dimension reduction treatment with a principal component.

Type I was oxidized melted tallow samples at different temperatures, which mainly contained five kinds of samples at 120 °C, 130 °C, 140 °C, 150 °C, and 160 °C. The content of 1-pentanal (almond), 1-hexanal (green, grassy), *(E)*-2-pentenal (fruity), 1-heptanal (fatty), 1-octanal (fatty), *(E)*-2-heptenal (fatty), 1-nonanal (fresh), *(E)*-2-octenal (cucumber), *(E)*-2-nonenal (cucumber), *(E)*-2-decenal (fatty), 2-undecenal (orange peel), and *(E,E)*-2,4-decadienal (fried) compounds were relatively high, and the characteristic odor of these compounds affected the overall odor of the five samples. The interaction between each odor caused each sample in Type I to, overall, present strong fruity and animal fat notes.

Type II samples mainly included 25 min, 35 min, 45 min, 55 min, and 65 min. The content of 1-pentanal (almond), *(E)*-2-decenal (fatty), *(E)*-2-heptenal (fatty), benzaldehyde (nutty), *(E,E)*-2,4-heptadienal (fatty), *(E,E)*-2,4-decadienal (fried), 1-octanal (fatty), 1-hexanal (green, grassy), *(E)*-2-pentenal (fresh), 1-heptanal (fatty), 2-tridecanone (creamy), 2-pentadecanone (floral), 2-octanone (fresh), 2-nonone (fresh), 2-decanone (orange peel), and 3-ethyl-cyclopentane-1,2-dione (caramel) were relatively high, and the characteristic odor of aldehyde and ketone compounds affected the overall odor of these five samples, it made the overall odor profiles of each sample in Type II show the odor of animal fat, muttony and sweet.

Type III mainly consisted of five samples at 18 L/h, 48 L/h, 78 L/h, 108 L/h, and 138 L/h. Compounds 3-methylbutyraldehyde (fresh), 1-pentanal (almond), 1-hexanal (green, grassy), *(E)*-2-pentenal (fruity), 1-heptanal (fatty), 1-octanal (fatty), *(E)*-2-nonenal (fatty), 1-nonanal (fresh), *(E)*-2-octenal (cucumber), *(E)*-2-nonenal (cucumber), *(E)*-2-decenal (fatty), α-pentylcinnamic aldehyde (floral), 3-methylbutyric acid (sweaty), 2- methylpropionic acid (fruity), hexanoic acid (cheese), and 2,6-ditertbutyl-4-methylphenol (camphor) and other compounds were relatively high. In addition, the characteristic odor of aldehyde and acid compounds affected the overall odor of the five samples, making each sample in Type III present a strong sweet and char flavor.

The proportion of aromatic compounds was analyzed by principal component analysis (PCA) based on the different factors of oxidized melted tallow sample by SPME-GC–MS. It was found that the principal components had significant differences among the different factors, but there was no significant difference at each level. Finally, according to the comprehensive evaluation of the oxidation index and sensory evaluation, the optimal reaction temperature was determined to be within the range of 140–160 °C, the reaction time was within the range of 25–45 min, and the airflow rate was within the range of 78–138 L/h.

### 2.6. Response Surface Optimization

Based on the single-factor test, three factors, including reaction temperature, reaction time, and airflow rate, were selected to design the response surface test. The factor-level table is shown in Table 6 and the response surface analysis scheme and results are shown in Table 7. The results were analyzed by Design-Expert data software and the response surface diagrams for the obtained data were shown in Figure 6, Figure 7 and Figure 8.

The significance test and analysis of variance for the regression equation are shown in Table 5. The significance for regression analysis of variance indicates that the regression is extremely significant, *p* = 0.0001, and non-significant, *p* = 1.0000, for the missing term, with R^2^ = 0.9726 and R^2^Adj = 0.9374. The model indicates that the equation fits the experiment well. When describing the relationship between each factor and response value, the linear relationship between the dependent variable and the independent variable is significant for the regression equation.

In the process of studying the influence of related variables on the processing results, in order to further study the interaction between variables, the quadratic regression model was analyzed by software and the response surface stereogram was drawn. The response surface isochronal diagram could intuitively reflect the effects of various factors on the response value to identify the optimal process parameters and the interaction between the parameters. The stereogram of the response surface showed that the fitting surface had the true minimum value. The center point of the smallest ellipse in the contour map was the lowest point of the response surface. The shape of the contour line could reflect the strength of the interaction effect. The ellipse indicated that the interaction between the two factors was significant, while the circle indicated the opposite. Figure 6, Figure 7 and Figure 8 show the effect of temperature, time, and airflow rate on sensory evaluation, and the contour plot shows that the influence of factors on sensory evaluation is significant.

After regression fitting of each factor, the regression equation was obtained as follows:

R1 = 9.03 − 0.020A − 0.28B − 0.53C + 0.000AB + 0.000AC + 0.000BC + 3.81BD − 0.26A^2^ − 0.42B^2^ − 0.75C^2^. The RSM analysis system showed that the optimal conditions for oxidative melting of tallow were a reaction temperature of 149.61 °C, a reaction time of 31.68 min, and an airflow rate of 97.44 L/h.

### 2.7. Accelerated Oxidation Test

After the determination of various oxidation indexes, sensory evaluation, and SPME-GC-O–MS analysis, the optimized oxidized melted tallow was subjected to an accelerated oxidation test, so as to ensure that various indexes of the produced oxidized tallow reached the standards.

The determination of each index in the accelerated oxidation test of oxidized melted tallow is shown in Figure 9. After being placed in an oven at 62 ± 1 °C for 15 d (the equivalent of 15 months), the AV value exceeded the standard after 13 d, and all indicators were qualified but not exceeding the standard before 12 d [19]. In addition, according to the analysis results of SPME-GC–MS, as shown in the Table 8, there was no significant difference in all substances of oxidized melted tallow in 0–15 d, such as the contents of acid compounds such as propionic acid, butanoic acid, octanoic acid, and decanoic acid, to confirm the quality of oxidized melted tallow.

## 3. Materials and Methods

### 3.1. Samples and Chemicals

Samples were provided by China Sichuan Guanghan Maidele Food Co., Ltd.

Raw tallow (200 g) was minced in a meat grinder, placed in a three-necked round-bottom flask, and placed in a 150 °C oil bath in which air was introduced at a rate of 108 L/h. The stirring speed was 200 r/min. The samples were smelled at 25 min, 35 min, 45 min, 55 min, and 65 min, respectively.

Ethyl ether, n-hexane, and anhydrous sodium sulfate, all having purities >99%, were purchased from the Lab Gou e-mall (Beijing, China). The compounds 2,4,5-Trimethylthiazole and n-alkanes (C_7_–C_30_) were provided by Sigma-Aldrich (St. Louis, MO, USA.). Air (99.999% purity) was obtained from Beijing Shuangquan lucky chance Industrial Gas Co., Ltd. and the liquid nitrogen was obtained from Xian Heyu Trading Co., Ltd. (Beijing, China).

### 3.2. Determination of Acid Value

The acid value of the sample was determined according to the GB 5009.229-2016 National Food Safety Standard [20].

### 3.3. Determination of Peroxide Value

The peroxide value of samples was determined according to the GB 5009.227-2016 National Food Safety Standard [21].

### 3.4. Determination of Anisidine Value

The anisidine value of samples was determined according to the GB/T 24304-2009 determination of the anisidine value of animal and vegetable fats and oils [22].

### 3.5. Analysis of Volatile Compounds

#### 3.5.1. Gas Chromatography-Olfactometry–Mass Spectrometry (GC-O–MS) Analysis

A GC–MS (7890A-7000, Agilent Technologies Inc., Santa Clara, CA, USA) instrument combined with an olfactory detection port (ODP4, Gerstel, Germany) was used to identify volatile odor compounds. Separation of odor-active substances in samples was performed on a polar DB-WAX capillary column (30 mm × 0.32 mm, 0.25 µm film thickness; J & W Scientific, Folsom, CA, USA). The gas chromatographic instrument condition includes an initial column temperature setting of 40 °C, holding for 3 min, followed by an increase in temperature up to 230 °C at 4 °C/min and holding for 3 min. Ultra-pure helium (99.999%, Beijing AP BAIF Gas Industry Co., Ltd., Beijing, China) was used as the carrier gas. The electron impact mass spectra were generated at an ionization energy of 70 eV with an *m/z* scan range of 25–370 amu. The temperatures of the mass spectrometer source and quadrupole were programmed at 230 °C and 150 °C, respectively. Moisture gas was delivered to the olfactory detection port through a blank capillary column.

#### 3.5.2. SPME-GC–MS Analysis of Volatile Compounds

The manual solid-phase microextraction technology was used to extract the volatile aroma compounds from the tallow. Five grams of the tallow sample and 1 μL of 2,4,5-Trimethylthiazole (concentration of 1.013 μg/μL, dissolved in n-hexane) were added into a 20 mL empty bottle as internal standard, mixed, and sealed. The sample was put in a constant-temperature water bath for 30 min, the temperature of the water bath was set at 60 °C, and the solid-phase microextraction fiber head (CAR/DVB/PDMS) was inserted for headspace adsorption for 30 min. After the adsorption, the SPME fiber was inserted into the GC inlet and desorpted at 250 °C for 5 min. Parameters of gas chromatography were as follows: the initial temperature was set at 40 °C in the heating program and the temperature was kept for 3 min. Firstly, the temperature was raised to 142 °C at 3.5 °C/min, then to 150 °C at 2 °C/min, then to 177 °C at 3.5 °C/min, and then to 200 °C at 6 °C/min. The carrier gas was helium, the constant flow rate was set at 1.2 mL/min, the temperature of the injection port was set at 250 °C, and the split ratio was set at splitless. The parameter of mass spectrometry were as follows: the ion source type was the electron bombardment, the electron energy was 70 eV, the transmission line temperature was 280 °C, the ion source temperature was maintained at 230 °C, the quadrupole temperature was set at 150 °C, the mass scanning range *m/z* was 40~250, and the solvent delay was 4 min.

#### 3.5.3. Qualitative and Quantitative Methods of Volatile Aroma Compounds

The qualitative methods of compounds specifically included matching mass spectra with the NIST 08 library to identify compounds and identifying compounds by the standardized compound retention index (RI) and smell (O).

The RI value is calculated according to the peak time of the target compound and the peak time of a series of alkane standards under the same gas quality parameters, and the formula is as follows:$$\begin{array}{l} RI=100n+\frac{100(t_{a}-t_{n})}{t_{n+1}-t_{n}} \end{array}$$
where t_a_ in the formula represents the retention time of sample a and t_n_ represents the retention time of C_n_ in normal paraffin standard (the retention time of sample A is between two adjacent normal paraffin C_n_ and C_n + 1_) [23].

Combined with SPME-GC-O–MS technology, a human nose was employed as a detector to smell the odor compounds separated by chromatographic column at the same time, and the odor description was recorded and compared with the RI value, and odor characteristics reported in the literature.

#### 3.5.4. Qualitative Analysis

All the standards used in this study were diluted with n-hexane to a certain concentration. In this study, an internal standard was added to semi-quantitate the aroma compounds in tallow and the relative concentrations of all volatile organic compounds were calculated using 2,4,5-trimethylthiazole as an internal standard. The concentration of each compound was calculated using the following formula:$$C_{i}=\frac{C_{is}\times A_{j}}{A_{is}}$$
where *C_i_* is the concentration of the compound, *C_is_* is the internal standard concentration of 1.013 μg/μL, *A_j_* is the chromatographic peak area of the compound, and *A_is_* is the chromatographic peak area of the internal standard.

### 3.6. Sensory Analysis

Sensory evaluation of oxidized melted tallow under different conditions, Samples of 7.5 g were weighted into an empty vial, put in a 60 °C water bath, and the smell after melting was evaluated. The sensory evaluation team is composed of 13 laboratory professionals. Sensory evaluation is conducted on eight aroma types, such as milky, muttony, animal fat, sweet, burnt, sour, fruit, and spoiled. The scores are from light to strong, with no smell at all being 0, a slight smell but not lasting being 1–2, and a strong smell being 3–4. The eight aroma types were weighted (Table 9), and the weighted total score of the sensory evaluation of each sample was calculated.

### 3.7. Accelerated Oxidation Analysis of Oil

This study used an oven heating accelerated oxidation test (the Schaal oven test). Different oil samples were placed in a colorless cover and put in a thermostat at 62 ± 1 °C for continuous heating and oxidation for 30 days and treated every 12 h [24].

Shook them and changed their positions in the thermostat at will. The acid value (AV), peroxide value (PV), and anisidine value (P-AV) were measured at 0, 3, 6, 9, 12, and 15 days after oxidation in order to understand the formation kinetics of oxidation products of the above-mentioned oils and fats.

### 3.8. Statistical Analysis

The obtained data were analyzed by one-way ANOVA with SPSS 13.0 (SPSS Inc, Chicago, IL, USA) software and *p* < 0.05 was significant. All data are expressed as mean standard deviation (SD, n = 3). Response surface optimization was analyzed by Design-Expert 8.0.6. PCA analysis was made by SIMCA-P + 11 software.

## 4. Conclusions

In this study, a qualitative and quantitative analysis of compounds with aroma characteristics was performed using SPME-GC–MS. A total of 51 aroma compounds were detected in melted tallow under different conditions. Through the determination of various indicators, sensory evaluation, and SPME-GC-O–MS analysis, the single factor range of the oxidation conditions of melted tallow was determined to be a reaction temperature within the range of 140–160 °C and a reaction time within the range of 25 min–45 min. The airflow rate was within the range of 78–138 L/h. The response surface methodology was used to determine the optimal process parameters for oxidative melting of tallow as follows: a reaction temperature of 149.61 °C, a reaction time of 31.68 min, and an airflow rate of 97.44 L/h. The accelerated oxidation test further verified the quality of oxidized tallow. In addition, because the aroma of oxidized beef tallow was complex and its flavor was unique, it was necessary to carry out the research on frying beef tallow condiments on this basis to further determine the flavor analysis of strong flavor beef tallow and the flavor performance during the actual application of beef tallow condiments. These studies would be conducive to the comprehensive analysis of the aroma characteristics and taste of a beef tallow hot pot.

## Figures and Tables

**Figure 1 molecules-27-09047-f001:**
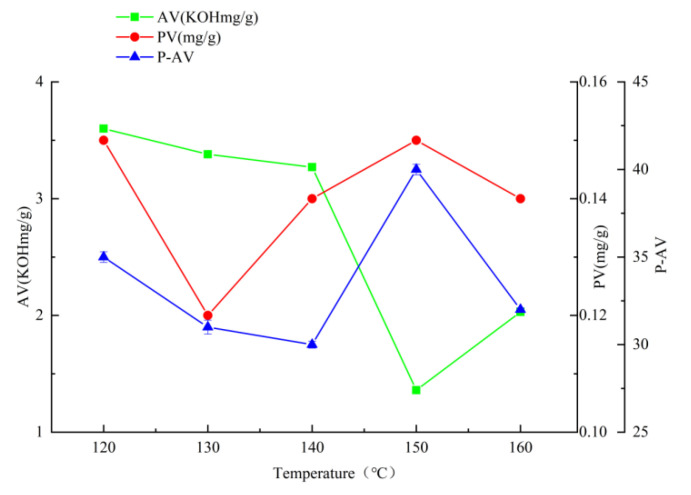
The effect of reaction temperature on AV, PV, and P-AV.

**Figure 2 molecules-27-09047-f002:**
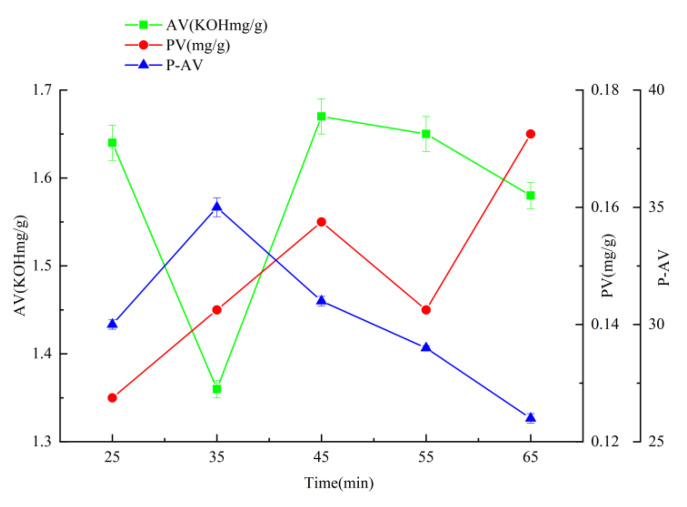
The effect of reaction time on AV, PV, and P-AV.

**Figure 3 molecules-27-09047-f003:**
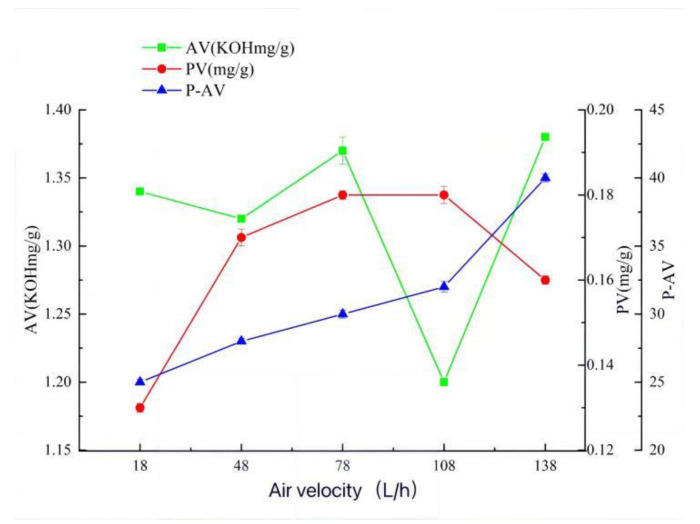
The effect of air velocity on AV, PV, and P-AV.

**Figure 4 molecules-27-09047-f004:**
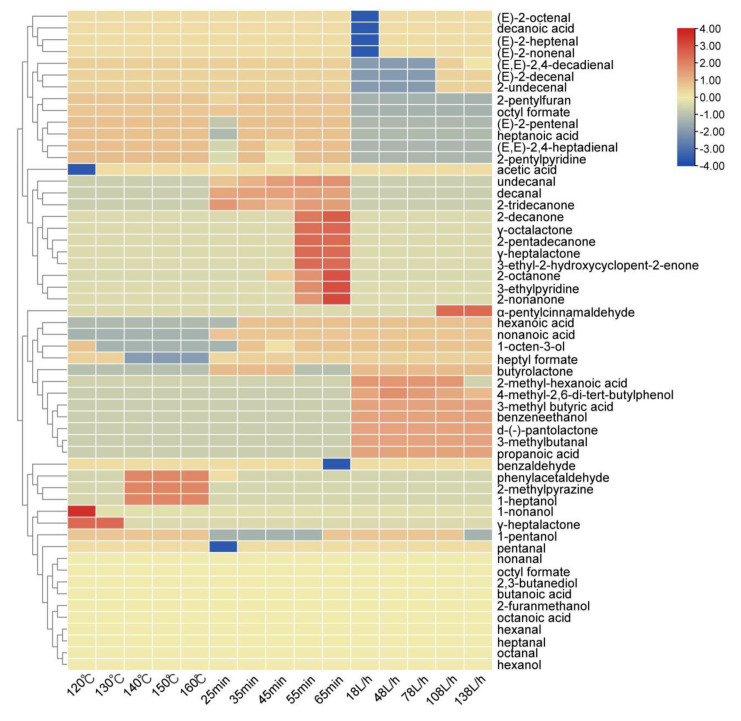
A content thermogram of aroma compounds in oxidized melted tallow under different conditions.

**Figure 5 molecules-27-09047-f005:**
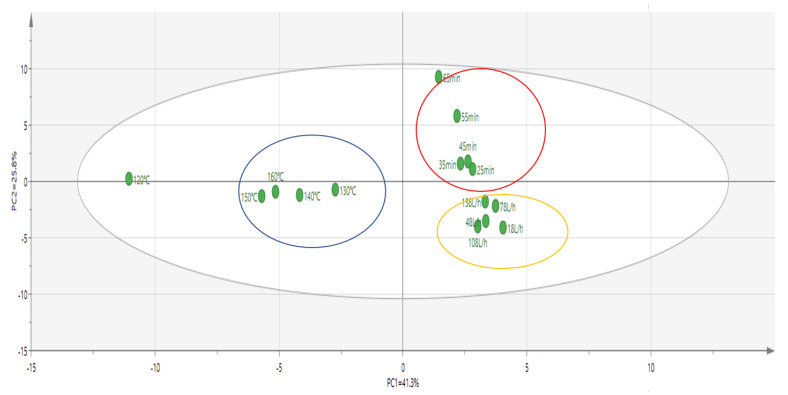
A SPME−GC−MS analysis of different factors of oxidized melted tallow samples, and principal component analysis of aroma compound proportion.

**Figure 6 molecules-27-09047-f006:**
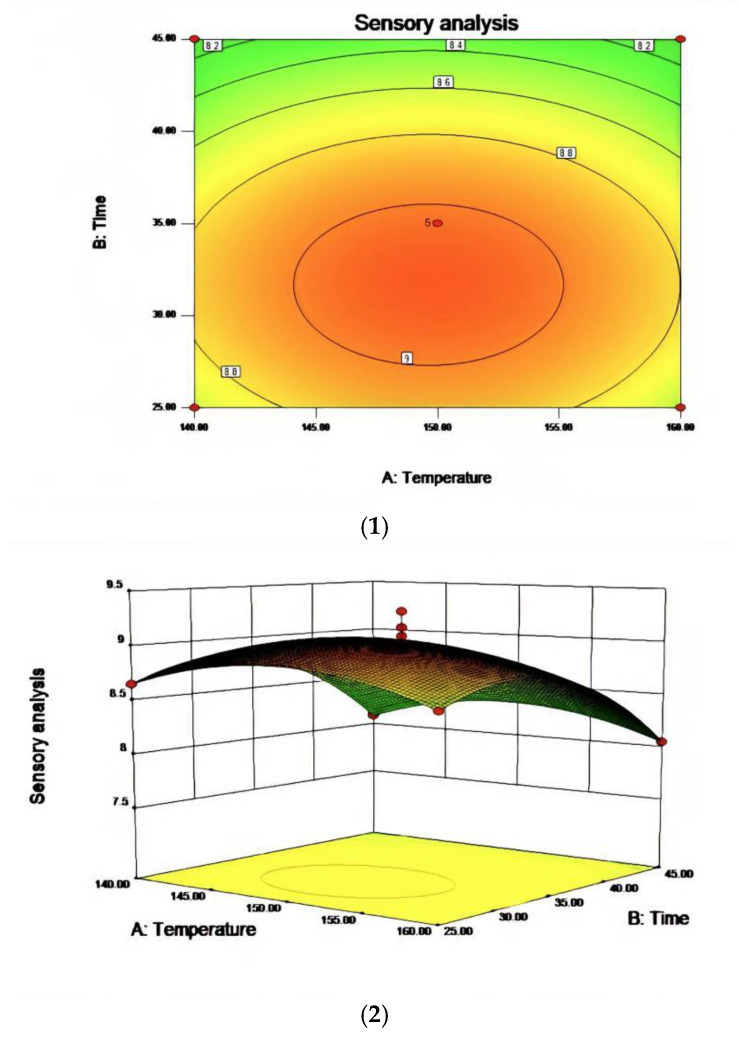
The effect of the interaction between reaction temperature and reaction time on sensory evaluation (**1**) and a contour map of the reaction temperature and reaction time (**2**).

**Figure 7 molecules-27-09047-f007:**
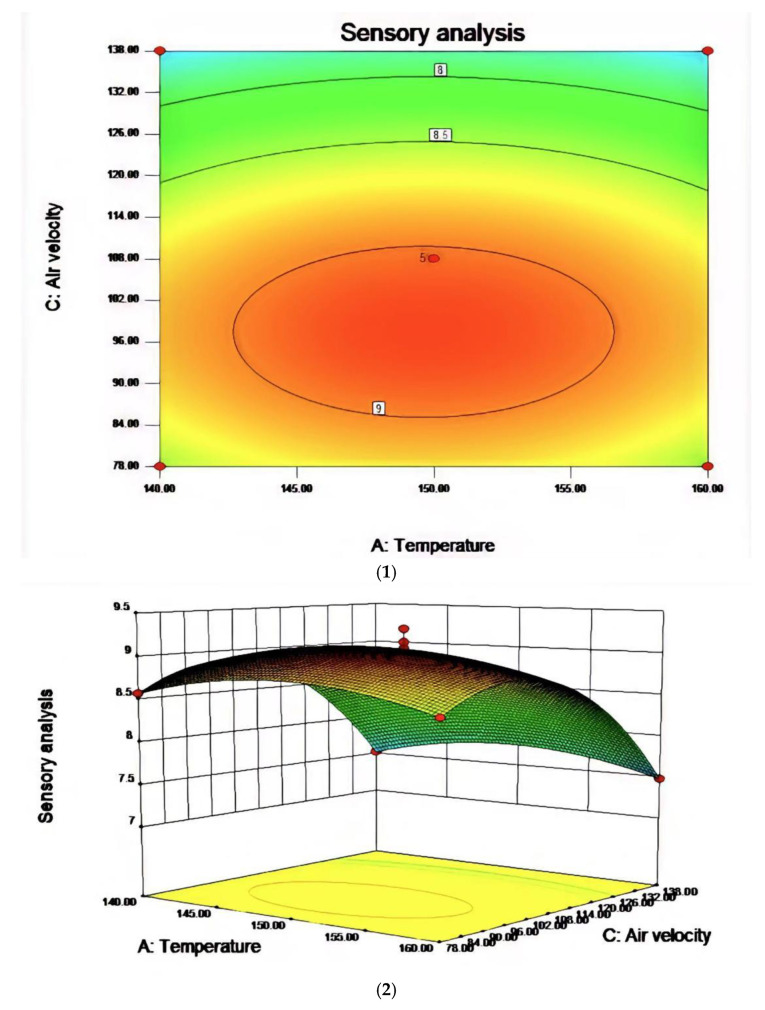
The effect of the interaction between reaction temperature and air velocity on sensory evaluation (**1**) and the contour map of reaction temperature and air velocity (**2**).

**Figure 8 molecules-27-09047-f008:**
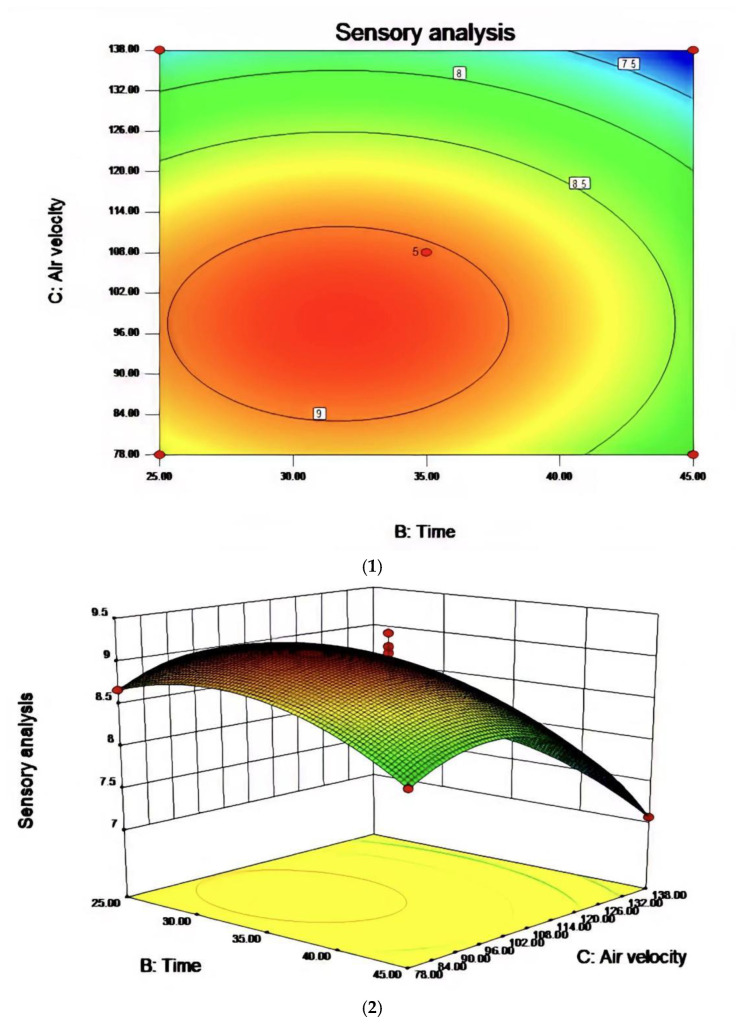
The effect of the interaction between reaction time and air velocity on sensory evaluation (**1**) and the contour map of reaction time and air velocity (**2**).

**Figure 9 molecules-27-09047-f009:**
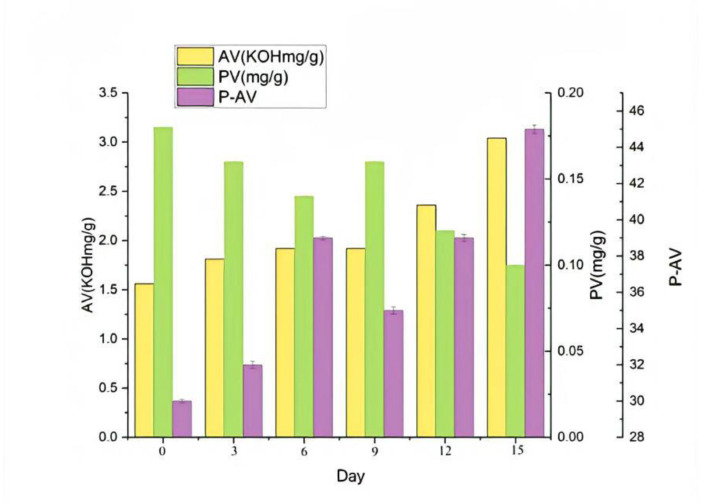
The test index chart of accelerated oxidation of oxidized tallow.

**Table 1 molecules-27-09047-t001:** Sensory score results of oxidized melted tallow under different conditions.

Different Reaction Conditions	Sample Number	Milky	Muttony	Animal Fat	Sweet	Burnt	Fruit	Sour	Spoiled	Weighted Total Score
Temperature	120 °C	0.6	0.34	0.64	0.48	0.2	0.12	−0.1	−0.14	2.14
	130 °C	0.72	0.3	0.72	0.48	0.2	0.12	−0.1	−0.2	2.24
	140 °C	0.78	0.34	0.8	0.48	0.2	0.12	−0.16	−0.16	2.4
	150 °C	0.94	0.24	0.68	0.58	0.1	0.24	−0.1	−0.14	2.54
	160 °C	0.64	0.34	0.9	0.48	0.24	0.12	−0.22	−0.14	2.36
Time	25 min	0.87	0.62	0.84	0.74	0.22	0.22	−0.25	−0.12	3.13
	35 min	0.85	0.65	0.89	0.71	0.20	0.27	−0.22	−0.06	3.29
	45 min	0.68	0.67	0.82	0.52	0.11	0.26	−0.35	−0.15	2.57
	55 min	0.65	0.73	0.65	0.55	0.11	0.21	−0.45	−0.16	2.28
	65 min	0.55	0.78	0.58	0.46	0.15	0.25	−0.69	−0.25	1.83
Air velocity	18 L/h	0.72	0.56	0.76	0.90	0.24	0.22	−0.24	−0.16	3.00
	48 L/h	0.90	0.52	0.80	0.72	0.24	0.22	−0.28	−0.14	2.98
	78 L/h	0.84	0.64	0.64	0.54	0.32	0.24	−0.24	−0.10	2.88
	108 L/h	0.84	0.68	0.72	0.84	0.24	0.22	−0.28	−0.14	3.12
	138 L/h	0.72	0.52	0.68	0.54	0.24	0.24	−0.24	−0.16	1.83

**Table 2 molecules-27-09047-t002:** SPME-GC-O–MS results of oxidized melted tallow at different temperatures.

	Compound Information				Relative Content (ng/g) ^y^				
No.	CAS	Component	RI ^x^	Odor	120 °C	130 °C	140 °C	150 °C	160 °C
		Aldehydes							
1	110-62-3	1-pentanal	962	almond	72.27 ± 12.26 ^a^	36.91 ± 1.23 ^b^	33.63 ± 1.28 ^b^	35.22 ± 1.65 ^b^	28.91 ± 2.36 ^b^
2	66-25-1	1-hexanal	1063	grassy	153.27 ± 24.55 ^a^	97.36 ± 2.85 ^b^	68.75 ± 1.79 ^c^	102.44 ± 4.27 ^b^	88.99 ± 8.96 ^bc^
3	111-71-7	*(E)*-2-pentenal	1128	fruity	149.27 ± 25.83 ^a^	49.93 ± 2.93 ^b^	34.39 ± 0.34 ^b^	35.92 ± 1.39 ^b^	42.20 ± 4.36 ^b^
4	6728-26-3	1-heptanal	1182	fatty	293.27 ± 51.27 ^a^	154.71 ± 6.97 ^b^	129.87 ± 5.25 ^b^	263.81 ± 0.38 ^a^	173.38 ± 19.54 ^b^
5	124-13-0	1-octanal	1274	fatty	375.27 ± 46.59 ^a^	151.56 ± 3.38 ^c^	144.96 ± 4.71 ^c^	233.31 ± 8.66 ^b^	264.61 ± 12.81 ^b^
6	18829-55-5	*(E)*-2-heptenal	1309	fatty	405.27 ± 59.55 ^a^	185.16 ± 5.86 ^bc^	146.09 ± 3.75 ^c^	195.24 ± 8.71 ^bc^	230.05 ± 11.42 ^b^
7	124-19-6	1-nonanal	1378	fresh	1020.27 ± 164.22 ^a^	407.17 ± 9.12 ^c^	364.92 ± 16.81 ^c^	683.57 ± 22.46 ^b^	558.30 ± 13.32 ^b^
8	2548-87-0	*(E)*-2-octenal	1413	cucumber	390.27 ± 51.58 ^a^	112.61 ± 9.53 ^c^	135.40 ± 2.90 ^c^	138.41 ± 8.27 ^c^	209.39 ± 12.43 ^b^
9	4313-03-5	*(E,E)*-2,4-heptadienal	1478	fatty	248.27 ± 22.28 ^a^	56.05 ± 2.86 ^d^	80.43 ± 5.46 ^c^	87.18 ± 5.40 ^c^	129.88 ± 2.83 ^b^
10	100-52-7	benzaldehyde	1504	nutty	30.27 ± 7.16 ^c^	33.00 ± 1.51 ^c^	43.74 ± 4.08 ^ab^	51.13 ± 2.35 ^a^	35.63 ± 2.76 ^bc^
11	18829-56-6	*(E)*-2-nonenal	1518	cucumber	633.27 ± 57.08 ^a^	170.32 ± 7.52 ^c^	193.24 ± 9.74 ^c^	164.58 ± 7.13 ^c^	247.69 ± 9.07 ^b^
12	122-78-1	phenylacetaldehyde	1648	sweet	ND	ND	72.32 ± 22.06 ^a^	46.62 ± 17.78 ^a^	60.28 ± 5.31 ^a^
13	3913-81-3	*(E)*-2-decenal	1639	fatty	1551.27 ± 56.24 ^a^	315.49 ± 15.51 ^c^	412.00 ± 21.29 ^b^	297.09 ± 6.37 ^c^	450.41 ± 6.62 ^b^
14	2463-77-6	2-undecenal	1730	orange peel	1106.27 ± 54.60 ^a^	220.72 ± 6.17 ^bc^	271.66 ± 28.44 ^b^	172.18 ± 9.53 ^c^	222.23 ± 6.50 ^bc^
15	25152-84-5	*(E,E)*-2,4-decadienal	1744	fried	167.27 ± 3.60 ^b^	10.10 ± 2.05 ^c^	199.87 ± 34.57 ^a^	126.04 ± 8.92 ^c^	205.27 ± 4.74 ^a^
		Alcohols							
16	71-41-0	1-pentanol	1244	spicy	63.27 ± 6.64 ^a^	38.15 ± 1.16 ^b^	38.56 ± 5.72 ^b^	56.06 ± 0.99 ^a^	35.30 ± 3.50 ^b^
17	3391-86-4	1-octen-3-ol	1430	mushroom	153.27 ± 23.18	ND	ND	ND	ND
18	111-70-6	1-heptanol	1447	floral	ND	ND	56.80 ± 0.71 ^b^	71.34 ± 3.39 ^a^	71.09 ± 4.46 ^a^
19	143-08-8	1-nonanol	1640	fatty	22.27 ± 7.98	ND	ND	ND	ND
		Sours							
20	64-19-7	acetic acid	1440	sour	ND	101.39 ± 2.62 ^c^	345.80 ± 4.57 ^a^	107.82 ± 5.67 ^c^	200.34 ± 1.65 ^b^
21	111-14-8	heptanoic acid	1940	sweaty	57.27 ± 7.61 ^c^	31.82 ± 4.87 ^d^	118.43 ± 2.43 ^b^	173.22 ± 20.81 ^a^	106.00 ± 8.79 ^b^
22	124-07-2	octanoic acid	2051	rot	49.27 ± 5.10 ^d^	27.86 ± 24.34 ^cd^	101.56 ± 1.00 ^b^	124.82 ± 8.78 ^a^	68.79 ± 5.47 ^c^
23	334-48-5	decanoic acid	2262	putrid	62.27 ± 8.84 ^b^	151.38 ± 12.82 ^b^	152.97 ± 37.80 ^a^	137.39 ± 16.62 ^a^	78.87 ± 3.31 ^a^
		Esters							
24	112-23-2	heptyl formate	1448	cucumber	145.27 ± 16.90	56.40 ± 5.43	ND	ND	ND
25	112-32-3	octyl formate	1548	cucumber	190.27 ± 17.70 ^a^	62.97 ± 1.36 ^c^	59.75 ± 1.88 ^c^	74.95 ± 3.60 ^bc^	81.09 ± 2.41 ^b^
26	105-21-5	γ-heptalactone	1784	coconut	29.27 ± 4.02	96.75 ± 1.60	ND	ND	ND
		Heterocycles							
27	3777-69-3	2-pentylfuran	1215	fruity	81.27 ± 12.24 ^a^	32.42 ± 1.65 ^c^	33.24 ± 1.36 ^c^	59.48 ± 2.11 ^b^	64.06 ± 7.48 ^b^
28	109-08-0	2-methylpyrazine	1256	nutty	ND	ND	5.87 ± 0.64 ^c^	18.42 ± 1.65 ^a^	11.98 ± 1.72 ^b^
29	2294-76-0	2-pentylpyridine	1600	fatty	66.27 ± 1.40 ^a^	21.17 ± 1.68 ^d^	31.85 ± 2.55 ^c^	44.89 ± 2.91 ^b^	35.50 ± 2.42 ^c^

x: RI, the retention index on capillaries DB-WAX. ND: not detected. y: the relative concentration stated as the mean ± SD and the unit is ppt (part per trillion, ng/g). ^a,b,c,d^: Different letters represent significant data, while the same letter represents weak significance.

**Table 3 molecules-27-09047-t003:** SPME-GC-O–MS results of oxidized melted tallow at different times.

	Compound Information				Relative Content (ng/g) y				
No.	CAS	Component	RI x	Odor	25 Min	35 Min	45 Min	55 Min	65 Min
		Aldehydes							
1	110-62-3	1-pentanal	962	almond	3.72 ± 0.78 ^e^	14.18 ± 0.25 ^c^	9.29 ± 1.91 ^d^	22.13 ± 2.09 ^b^	33.17 ± 0.51 ^a^
2	66-25-1	1-hexanal	1063	grassy	12.23 ± 2.76 ^d^	45.66 ± 0.59 ^b^	31.02 ± 1.53 ^c^	40.79 ± 11.90 ^bc^	68.54 ± 2.98 ^a^
3	111-71-7	*(E)*-2-pentenal	1128	fruity	0.79 ± 0.17 ^d^	11.46 ± 1.02 ^b^	6.97 ± 2.65 ^c^	14.55 ± 6.88 ^bc^	30.46 ± 6.62 ^a^
4	6728-26-3	1-heptanal	1182	fatty	11.78 ± 7.72 ^c^	36.79 ± 4.01 ^b^	33.17 ± 12.02 ^b^	36.62 ± 6.19 ^b^	86.38 ± 1.41 ^a^
5	124-13-0	1-octanal	1274	fatty	17.09 ± 0.71 ^d^	35.96 ± 0.25 ^c^	24.47 ± 2.11 ^d^	53.40 ± 4.90 ^b^	82.43 ± 6.45 ^a^
6	18829-55-5	*(E)*-2-heptenal	1309	fatty	16.39 ± 3.57 ^e^	65.45 ± 0.17 ^c^	41.97 ± 2.50 ^d^	103.67 ± 1.78 ^b^	143.25 ± 0.97 ^a^
7	124-19-6	nonanal	1378	fresh	36.64 ± 2.22 ^e^	97.19 ± 0.97 ^c^	64.21 ± 8.54 ^d^	143.24 ± 1.89 ^b^	201.25 ± 6.89 ^a^
8	2548-87-0	*(E)*-2-octenal	1413	cucumber	8.31 ± 0.64 ^e^	27.06 ± 0.77 ^c^	17.72 ± 0.67 ^d^	53.35 ± 1.07 ^b^	87.75 ± 1.18 ^a^
9	4313-03-5	*(E,E)*-2,4-heptadienal	1478	fatty	1.41 ± 0.60 ^d^	6.28 ± 1.66 ^c^	3.11 ± 0.89 ^d^	27.60 ± 0.93 ^b^	57.63 ± 2.38 ^a^
10	112-31-2	1-decanal	1472	fatty	3.90 ± 1.37 ^a^	5.66 ± 0.74 ^a^	4.57 ± 1.93 ^a^	5.81 ± 1.04 ^a^	5.79 ± 0.97 ^a^
11	100-52-7	benzaldehyde	1504	nutty	6.04 ± 1.41 ^a^	6.66 ± 0.53 ^a^	6.10 ± 1.62 ^a^	5.85 ± 0.03 ^a^	ND
12	18829-56-6	*(E)*-2-nonenal	1518	cucumber	14.71 ± 3.76 ^d^	35.54 ± 3.05 ^c^	29.07 ± 5.13 ^c^	77.91 ± 8.31 ^b^	145.95 ± 1.46 ^a^
13	112-44-7	undecanal	1622	soap	2.32 ± 1.39 ^c^	2.91 ± 0.37 ^bc^	3.58 ± 1.95 ^bc^	5.14 ± 0.74 ^ab^	6.71 ± 0.52 ^a^
14	122-78-1	phenylacetaldehyde	1648	sweet	1.24 ± 0.69	ND	ND	ND	ND
15	3913-81-3	*(E)*-2-decenal	1639	fatty	32.13 ± 1.81 ^d^	74.34 ± 5.01 ^c^	64.89 ± 1.85 ^c^	193.20 ± 10.55 ^b^	344.06 ± 8.67 ^a^
16	2463-77-6	2-undecenal	1730	orange peel	21.39 ± 1.69 ^d^	68.24 ± 1.61 ^c^	44.64 ± 2.39 ^cd^	140.63 ± 8.83 ^b^	259.58 ± 29.81 ^a^
17	25152-84-5	*(E,E)*-2,4-decadienal	1744	fried	5.95 ± 0.29 ^c^	22.08 ± 1.99 ^c^	11.03 ± 0.67 ^c^	52.01 ± 4.59 ^b^	124.47 ± 18.70 ^a^
		Sours							
18	64-19-7	acetic acid	1440	sour	43.85 ± 2.57 ^a^	43.11 ± 1.14 ^a^	21.63 ± 0.94 ^b^	20.49 ± 0.45 ^b^	20.84 ± 0.69 ^b^
19	142-62-1	hexanoic acid	1836	cheese	ND	10.66 ± 0.32 ^bc^	7.74 ± 0.43 ^c^	12.91 ± 3.92 ^b^	26.07 ± 1.68 ^a^
20	111-14-8	heptanoic acid	1940	sweaty	ND	6.45 ± 0.86 ^c^	5.25 ± 1.13 ^bc^	10.02 ± 3.44 ^b^	16.45 ± 2.08 ^a^
21	124-07-2	octanoic acid	2051	rot	17.87 ± 0.94 ^c^	19.73 ± 2.09 ^bc^	18.18 ± 0.98 ^bc^	22.92 ± 4.42 ^b^	31.51 ± 0.93 ^a^
22	112-05-0	nonanoic acid	2144	waxy	448.77 ± 2.99 ^b^	709.50 ± 160.53 ^a^	623.77 ± 52.50 ^ab^	664.24 ± 63.45 ^a^	814.94 ± 31.35 ^a^
23	334-48-5	decanoic acid	2262	putrid	6.62 ± 0.14 ^a^	12.32 ± 1.83 ^a^	9.89 ± 2.42 ^a^	12.30 ± 5.51 ^a^	9.50 ± 2.27 ^a^
		Ketones							
24	111-13-7	2-octanone	1271	earthy	ND	ND	0.71 ± 0.64 ^a^	1.48 ± 0.94 ^a^	2.33 ± 0.89 ^a^
25	821-55-6	2-nonanone	1389	soap	ND	ND	ND	1.33 ± 0.36	2.39 ± 1.21
26	693-54-9	2-decanone	1482	citrus	ND	ND	ND	2.90 ± 1.16	3.54 ± 1.71
27	593-08-8	2-tridecanone	1814	creamy	2.81 ± 0.61 ^a^	2.35 ± 0.69 ^a^	2.05 ± 1.18 ^a^	2.67 ± 0.97 ^a^	2.64 ± 0.30 ^a^
28	21835-01-8	3-ethyl-2-hydroxycyclopent-2-enone	1894	caramel	ND	ND	ND	4.30 ± 2.60	10.67 ± 0.18
29	2345-28-0	2-pentadecanone	2021	jasmine	ND	ND	ND	9.43 ± 1.27	12.22 ± 1.68
		Esters							
30	112-23-2	heptyl formate	1448	cucumber	3.79 ± 0.48 ^d^	11.92 ± 1.10 ^bc^	6.63 ± 0.52 ^cd^	16.58 ± 8.75 ^b^	35.14 ± 0.71 ^a^
31	112-32-3	octyl formate	1548	cucumber	7.90 ± 0.44 ^c^	13.03 ± 0.30 ^bc^	8.60 ± 0.93 ^c^	22.14 ± 1.63 ^b^	35.11 ± 2.09 ^a^
32	96-48-0	γ-butyrolactone	1618	peach	8.07 ± 1.85 ^a^	7.16 ± 0.48 ^a^	5.55 ± 2.28 ^a^	ND	ND
33	105-21-5	γ-heptalactone	1784	coconut	ND	ND	ND	5.20 ± 1.64	14.71 ± 0.26
34	104-50-7	γ-octalactone	1898	coconut	ND	ND	ND	3.81 ± 1.55	9.67 ± 0.60
		Alcohols							
35	71-41-0	1-pentanol	1244	spicy	ND	ND	ND	ND	23.20 ± 1.28
36	3391-86-4	1-octen-3-ol	1430	mushroom	ND	4.94 ± 0.30 ^bc^	2.93 ± 0.12 ^c^	5.80 ± 2.44 ^b^	10.42 ± 0.29 ^a^
		Heterocycles							
37	3777-69-3	2-pentylfuran	1215	fruity	3.50 ± 0.25 ^c^	5.72 ± 0.52 ^bc^	4.70 ± 0.35 ^bc^	7.12 ± 2.65 ^b^	15.52 ± 0.08 ^a^
38	536-78-7	3-ethylpyridine	1387	hazelnut	ND	ND	ND	2.58 ± 1.09	4.96 ± 0.56
39	2294-76-0	2-pentylpyridine	1600	fatty	1.60 ± 0.16 ^c^	4.91 ± 0.39 ^b^	2.11 ± 0.29 ^c^	4.07±1.60 ^b^	7.32±0.61 ^a^

x: RI, the retention index on capillaries DB-WAX. ND: not detected. y: the relative concentration stated as the mean ± SD and the unit is ppt (part per trillion, ng/g).^a,b,c,d,e^: Different letters represent significant data, while the same letter represents weak significance.

**Table 4 molecules-27-09047-t004:** SPME-GC-O–MS results of oxidized melted tallow at different airflow rates.

	Compound Information				Relative Content (ng/g)^y^				
No.	CAS	Component	RI ^x^	Odor	18 L/h	48 L/h	78 L/h	108 L/h	138 L/h
		Aldehydes							
1	590-86-3	3-methylbutanal	941	earthy	32.37 ± 2.37 ^a^	33.60 ± 3.96 ^a^	21.59 ± 0.31 ^bc^	31.09 ± 0.03 ^ab^	14.60 ± 0.96 ^c^
2	110-62-3	1-pentanal	962	almond	30.53 ± 7.07 ^b^	28.05 ± 5.48 ^b^	11.90 ± 0.79 ^c^	45.89 ± 9.92 ^a^	9.78 ± 1.08 ^c^
3	66-25-1	1-hexanal	1063	grassy	85.38 ± 14.69 ^bc^	112.29 ± 24.13 ^b^	53.94 ± 10.72 ^c^	186.10 ± 34.37 ^a^	46.58 ± 6.24 ^c^
4	111-71-7	1-heptanal	1182	fatty	40.90 ± 3.83 ^c^	88.09 ± 21.79 ^b^	22.66 ± 4.08 ^c^	118.91 ± 9.93 ^a^	28.81 ± 2.06 ^c^
5	124-13-0	1-octanal	1274	fatty	21.81 ± 1.19 ^c^	40.42 ± 11.10 ^b^	11.41 ± 0.60 ^c^	62.87 ± 5.90 ^a^	22.60 ± 2.23 ^c^
6	18829-55-5	*(E)*-2-heptenal	1309	fatty	ND	23.97 ± 5.34 ^b^	14.01 ± 0.04 ^d^	41.31 ± 1.45 ^a^	21.30 ± 1.54 ^c^
7	124-19-6	1-nonanal	1378	fresh	33.78 ± 8.71 ^c^	60.76 ± 0.07 ^b^	29.24 ± 0.76 ^c^	112.82 ± 4.04 ^a^	55.97 ± 5.05 ^b^
8	2548-87-0	*(E)*-2-octenal	1413	cucumber	2.86 ± 0.64 ^d^	7.27 ± 1.77 ^b^	5.40 ± 2.49 ^c^	14.71 ± 0.37 ^a^	7.53 ± 0.68 ^b^
9	100-52-7	benzaldehyde	1504	nutty	20.66 ± 6.36 ^a^	9.94 ± 4.68 ^b^	6.11 ± 3.31 ^b^	9.02 ± 0.36 ^b^	5.97 ± 0.77 ^b^
10	18829-56-6	*(E)*-2-nonenal	1518	cucumber	ND	7.88 ± 0.26 ^b^	8.01 ± 2.24 ^b^	18.68 ± 1.20 ^a^	9.21 ± 1.91 ^b^
11	3913-81-3	*(E)*-2-decenal	1639	fatty	ND	ND	ND	20.57 ± 2.17	22.21 ± 6.56
12	2463-77-6	2-undecenal	1730	orange peel	ND	ND	ND	4.66 ± 3.59	7.37 ± 1.38
13	25152-84-5	*(E,E)*-2,4-decadienal	1744	fried	ND	ND	ND	5.11 ± 0.28	3.32 ± 0.18
14	122-40-7	α-pentylcinnamaldehyde	2245	floral	ND	ND	ND	30.97 ± 9.81	23.82 ± 0.73
		Sours							
15	64-19-7	acetic acid	1440	sour	110.03 ± 5.23 ^b^	177.73 ± 10.71 ^a^	63.92 ± 1.65 ^d^	177.18 ± 13.18 ^a^	82.10 ± 5.34 ^c^
16	79-09-4	propanoic acid	1529	sour	14.51 ± 0.87 ^c^	25.38 ± 2.92 ^a^	12.55 ± 2.76 ^c^	20.55 ± 1.16 ^b^	11.99 ± 3.36 ^c^
17	503-74-2	3-methyl butyric acid	1680	sweaty	109.05 ± 2.33 ^a^	42.32 ± 2.84 ^c^	30.63 ± 1.03 ^d^	60.72 ± 10.95 ^b^	10.82 ± 0.95 ^e^
18	79-31-2	2-methyl-hexanoic acid	1700	fruity	128.79 ± 2.10 ^a^	36.48 ± 2.76 ^c^	38.40 ± 0.71 ^c^	59.17 ± 8.99 ^b^	ND
19	142-62-1	hexanoic acid	1836	cheese	9.81 ± 2.39 ^b^	7.16 ± 1.49 ^b^	13.41 ± 1.83 ^a^	15.11 ± 1.22 ^a^	7.04 ± 0.13 ^b^
20	124-07-2	octanoic acid	2051	rot	7.20 ± 3.07 ^c^	5.43 ± 0.16 ^d^	4.04 ± 0.72 ^e^	7.87 ± 2.23 ^b^	9.69 ± 0.56 ^a^
21	112-05-0	nonanoic acid	2144	waxy	6.74 ± 1.96 ^c^	19.20 ± 4.20 ^ab^	7.43 ± 1.68 ^c^	17.46 ± 9.66 ^ab^	21.30 ± 5.90 ^a^
22	334-48-5	decanoic acid	2262	putrid	ND	11.62 ± 3.72 ^a^	5.91 ± 1.31 ^c^	5.31 ± 1.57 ^d^	7.69 ± 1.70 ^b^
		Alcohols							
23	71-41-0	1-pentanol	1244	spicy	35.32 ± 0.87 ^b^	36.11 ± 5.77 ^b^	12.84 ± 1.00 ^c^	52.92 ± 9.17 ^a^	ND
24	3391-86-4	1-octen-3-ol	1430	mushroom	6.83 ± 0.18 ^bc^	8.59 ± 1.00 ^b^	5.64 ± 0.73 ^cd^	15.42 ± 2.10 ^a^	3.99 ± 0.17 ^d^
25	60-12-8	benzeneethanol	1912	floral	28.71 ± 5.14 ^a^	12.20 ± 1.13 ^c^	17.66 ± 0.21 ^b^	19.99 ± 0.53 ^b^	9.58 ± 0.55 ^c^
		Esters							
26	112-23-2	heptyl formate	1448	cucumber	8.00 ± 2.10 ^c^	14.62 ± 1.88 ^b^	4.62 ± 1.13 ^d^	19.92 ± 2.29 ^a^	5.86 ± 0.78 ^d^
27	96-48-0	butyrolactone	1618	peach	16.54 ± 2.65 ^b^	23.01 ± 3.32 ^a^	7.56 ± 1.00 ^c^	18.14 ± 3.22 ^b^	7.95 ± 0.48 ^c^
28	599-04-2	d-(-)-pantolactone	1998	cotton candy	9.54 ± 1.03 ^a^	5.89 ± 0.24 ^cd^	6.76 ± 0.36 ^bc^	7.20 ± 0.61 ^b^	4.86 ± 0.91 ^d^
		Others							
29	128-37-0	4-methyl-2,6-di-tert-butylphenol	1989	camphor	2.99 ± 1.12 ^a^	3.62 ± 1.58 ^a^	3.26 ± 1.48 ^a^	2.94 ± 0.97 ^a^	2.39 ± 0.36 ^a^

x: RI, the retention index on capillaries DB-WAX. ND: not detected. y: the relative concentration stated as the mean ± SD and the unit is ppt (part per trillion, ng/g). ^a,b,c,d^: Different letters represent significant data, while the same letter represents weak significance.

**Table 5 molecules-27-09047-t005:** The proportion of aroma compounds in oxidized melted tallow under different conditions.

Condition	Number	Aldehyde	Alcohol	Ester	Sour	Ketone	Heterocycle	Other
Temperature	120 °C	56.00%	12.00%	12.00%	12.00%	0.00%	8.00%	0.00%
	130 °C	58.33%	4.17%	12.50%	16.67%	0.00%	8.33%	0.00%
	140 °C	60.00%	4.00%	4.00%	16.00%	0.00%	12.00%	0.00%
	150 °C	60.00%	4.00%	4.00%	16.00%	0.00%	12.00%	0.00%
	160 °C	60.00%	4.00%	4.00%	16.00%	0.00%	12.00%	0.00%
Time	25 min	59.26%	0.00%	11.11%	14.81%	3.70%	7.41%	0.00%
	35 min	51.72%	3.45%	10.34%	20.69%	3.45%	6.90%	0.00%
	45 min	50.00%	6.67%	10.00%	20.00%	6.67%	6.67%	0.00%
	55 min	41.67%	5.56%	11.11%	16.67%	16.67%	8.33%	0.00%
	65 min	38.89%	8.33%	11.11%	16.67%	16.67%	8.33%	0.00%
Air velocity	18 L/h	36.36%	13.64%	13.64%	31.82%	0.00%	0.00%	4.55%
	48 L/h	40.00%	12.00%	12.00%	32.00%	0.00%	0.00%	4.00%
	78 L/h	40.00%	12.00%	12.00%	32.00%	0.00%	0.00%	4.00%
	108 L/h	48.28%	10.34%	10.34%	27.59%	0.00%	0.00%	3.45%
	138 L/h	51.85%	7.41%	11.11%	25.93%	0.00%	0.00%	3.70%

**Table 6 molecules-27-09047-t006:** Factors and levels of the response surface analysis.

Level	Factor		
	A Temperature/ °C	B Time/min	C Air Velocity/L·h^−1^
1	140	25	78
0	150	35	108
−1	160	45	138

**Table 7 molecules-27-09047-t007:** Variance analysis of the response surface quadratic model.

Source	Sum of Squares	df	Mean Square	F-Value	*p*-Value Prob > F	Significance
Model	6.54	9	0.73	27.61	0.0001	significance
A: Temperature	0.0032	1	0.0032	0.12	0.7375	
B: Time	0.63	1	0.63	23.85	0.0018	
C: Air velocity	2.2	1	2.2	83.83	<0.0001	
AB	0	1	0	0	1	
AC	0	1	0	0	1	
BC	0	1	0	0	1	
A^2^	0.29	1	0.29	10.99	0.0129	
B^2^	0.75	1	0.75	28.51	0.0011	
C^2^	2.35	1	2.35	89.33	<0.0001	
Residual	0.18	7	0.026			
Lack of fit	0	3	0	0	1	no significance
Pure error	0.18	4	0.046			
Cor total	6.72	16				
R^2^ = 0.9726; R^2^Adj = 0.9374						

**Table 8 molecules-27-09047-t008:** SPME-GC-O–MS results of accelerated oxidation test of melted tallow.

	Compound Information				Relative Content (ng/g) ^y^					
No.	CAS	Component	RI ^x^	Odor	0 Day	3 Day	6 Day	9 Day	12 Day	15 Day
		Aldehydes								
1	590-86-3	3-methylbutanal	941	earthy	263.63 ± 24.09 ^b^	299.01 ± 21.64 ^ab^	308.86 ± 36.75 ^a^	207.17 ± 11.48 ^c^	177.67 ± 19.55 ^cd^	147.43 ± 0.94 ^d^
2	110-62-3	1-pentanal	962	almond	27.23 ± 4.30 ^c^	83.19 ± 12.63 ^ab^	91.50 ± 6.15 ^a^	92.22 ± 1.41 ^a^	73.89 ± 5.41 ^b^	94.86 ± 7.69 ^a^
3	66-25-1	1-hexanal	1063	grassy	217.41 ± 15.65 ^bc^	238.44 ± 20.54 ^bc^	259.96 ± 28.52 ^b^	217.62 ± 39.70 ^bc^	315.29 ± 12.72 ^a^	201.65 ± 1.88 ^c^
4	111-71-7	1-heptanal	1182	fatty	223.38 ± 38.72 ^b^	285.09 ± 14.72 ^b^	387.00 ± 43.87 ^a^	411.84 ± 33.15 ^a^	418.97 ± 40.68 ^a^	439.12 ± 2.26 ^a^
5	124-13-0	1-octanal	1274	fatty	161.51 ± 11.87 ^c^	205.33 ± 17.93 ^b^	264.49 ± 0.83 ^a^	292.47 ± 18.55 ^a^	272.33 ± 9.68 ^a^	291.44 ± 17.48 ^a^
6	124-19-6	1-nonanal	1378	fresh	85.31 ± 5.89 ^c^	149.65 ± 20.19 ^b^	124.06 ± 5.30 ^bc^	106.56 ± 15.58 ^c^	219.22 ± 13.82 ^a^	252.59 ± 26.67 ^a^
7	100-52-7	benzaldehyde	1504	nutty	71.66 ± 15.75 ^c^	67.19 ± 5.52 ^c^	70.21 ± 12.28 ^c^	51.73 ± 9.53 ^c^	114.50 ± 11.16 ^b^	148.00 ± 5.25 ^a^
8	18829-56-6	*(E)*-2-nonenal	1518	cucumber	ND	ND	ND	28.64 ± 3.19 ^b^	46.07 ± 2.09 ^a^	48.28 ± 4.05 ^a^
9	122-78-1	phenylacetaldehyde	1648	sweet	28.16 ± 2.13 ^bc^	31.95 ± 6.09 ^bc^	50.92 ± 8.44 ^a^	12.49 ± 0.49 ^a^	11.93 ± 0.98 ^b^	20.38 ± 6.25 ^c^
10	3913-81-3	*(E)*-2-decenal	1639	fatty	22.74 ± 3.15 ^abc^	11.87 ± 5.96 ^c^	22.45 ± 5.92 ^abc^	13.38 ± 4.43 ^bc^	26.40 ± 0.18 ^ab^	27.29 ± 8.83 ^a^
11	2463-77-6	2-undecenal	1730	orange peel	ND	14.92 ± 6.00 ^b^	12.84 ± 6.12 ^b^	13.75 ± 4.21 ^b^	16.32 ± 6.15 ^a^	13.54 ± 5.78 ^b^
		Sours								
12	64-19-7	acetic acid	1440	sour	1990.12 ± 276.50 ^ab^	2271.87 ± 95.35 ^a^	2082.05 ± 16.03 ^a^	2194.89 ± 61.30 ^a^	1654.51 ± 40.09 ^bc^	1528.08 ± 31.3 ^c^
13	79-09-4	propanoic acid	1529	sour	48.65 ± 9.81 ^b^	79.32 ± 6.34 ^a^	77.86 ± 5.21 ^a^	79.80 ± 6.32 ^a^	70.94 ± 11.79 ^a^	68.61 ± 6.18 ^a^
14	107-92-6	butanoic acid	1628	sour	32.84 ± 3.76 ^ab^	30.88 ± 2.64 ^ab^	32.27 ± 1.14 ^ab^	34.98 ± 3.49 ^a^	25.80 ± 1.00 ^b^	28.24 ± 4.52 ^ab^
15	503-74-2	3-methylbutyricacid	1680	sweaty	127.84 ± 4.28 ^bc^	168.12 ± 0.85 ^a^	146.98 ± 19.08 ^ab^	144.41 ± 4.81 ^abc^	128.59 ± 21.52 ^bc^	115.52 ± 1.67 ^c^
16	142-62-1	hexanoic acid	1836	cheese	23.69 ± 6.23 ^d^	37.02 ± 2.71 ^cd^	50.68 ± 9.91 ^bc^	68.71 ± 4.30 ^ab^	79.06 ± 13.98 ^a^	81.04 ± 6.54 ^a^
17	111-14-8	heptanoic acid	1940	sweaty	ND	14.30 ± 1.93 ^a^	15.81 ± 4.67 ^a^	14.18 ± 4.68 ^a^	21.99 ± 10.31 ^a^	19.11 ± 1.25 ^a^
18	124-07-2	octanoic acid	2051	rot	21.85 ± 5.97 ^a^	19.77 ± 4.76 ^a^	27.91 ± 5.92 ^a^	24.94 ± 6.66 ^a^	28.22 ± 1.90 ^a^	28.37 ± 3.64 ^a^
		Sours								
19	112-05-0	nonanoic acid	2144	waxy	17.86 ± 6.85 ^b^	66.92 ± 5.62 ^a^	56.05 ± 8.26 ^a^	21.78 ± 9.12 ^b^	25.71 ± 0.55 ^b^	17.31 ± 2.71 ^b^
20	334-48-5	decanoic acid	2262	putrid	21.21 ± 2.45 ^bc^	7.93 ± 2.58 ^d^	13.76 ± 3.79 ^cd^	17.47 ± 6.44 ^bcd^	31.86 ± 6.59 ^a^	24.81 ± 5.64 ^ab^
		Alcohols								
21	71-41-0	1-pentanol	1244	spicy	62.61 ± 8.12 ^b^	76.58 ± 7.52 ^a^	65.40 ± 2.83 ^b^	59.88 ± 2.95 ^b^	64.89 ± 0.98 ^b^	44.14 ± 4.34 ^c^
22	111-27-3	1-hexanol	1345	fresh	30.56 ± 5.78 ^a^	29.70 ± 7.76 ^a^	32.04 ± 5.09 ^a^	33.55 ± 0.27 ^a^	32.48 ± 8.31 ^a^	29.88 ± 0.37 ^a^
23	3391-86-4	1-octen-3-ol	1430	mushroom	11.10 ± 2.87 ^b^	15.66 ± 1.27 ^ab^	18.47 ± 4.01 ^ab^	18.97 ± 6.30 ^ab^	22.69 ± 3.47 ^a^	21.42 ± 1.17 ^a^
24	513-85-9	2,3-butanediol	1570	butter	36.02 ± 0.80 ^b^	89.88 ± 2.78 ^a^	81.31 ± 16.25 ^a^	79.44 ± 9.25 ^a^	50.38 ± 3.92 ^b^	53.84 ± 6.28 ^b^
25	60-12-8	benzeneethanol	1912	floral	35.38 ± 18.36 ^b^	45.98 ± 1.91 ^ab^	52.89 ± 0.41 ^a^	60.01 ± 1.10 ^a^	57.96 ± 6.15 ^a^	51.97 ± 5.38 ^ab^
		Esters								
26	112-23-2	heptyl formate	1448	cucumber	39.43 ± 4.76 ^b^	57.00 ± 3.50 ^ab^	57.40 ± 11.47 ^ab^	62.09 ± 12.97 ^a^	60.43 ± 12.68 ^ab^	52.36 ± 1.06 ^ab^
27	112-32-3	octyl formate	1548	cucumber	29.45 ± 11.21 ^a^	29.64 ± 3.49 ^a^	36.26 ± 12.20 ^a^	31.86 ± 5.65 ^a^	39.77 ± 1.08 ^a^	40.49 ± 4.83 ^a^
28	96-48-0	butyrolactone	1618	peach	69.99 ± 9.03 ^b^	89.14 ± 7.29 ^a^	83.63 ± 10.08 ^ab^	85.87 ± 6.07 ^a^	74.12 ± 6.86 ^ab^	70.37 ± 2.46 ^b^
29	599-04-2	d-(-)-pantolactone	1998	cotton candy	9.29 ± 0.45 ^b^	11.64 ± 0.50 ^ab^	11.37 ± 2.61 ^ab^	13.90 ± 2.30 ^a^	12.50 ± 1.04 ^ab^	11.58 ± 1.30 ^ab^
		Heterocyclics								
30	3777-69-3	2-pentylfuran	1215	fruity	ND	ND	14.09 ± 5.89 ^b^	25.42 ± 0.06 ^ab^	33.88 ± 7.46 ^a^	39.43 ± 12.59 ^a^
31	98-00-0	2-furanmethanol	1635	burnt	12.02 ± 2.35 ^b^	14.88 ± 0.37 ^a^	12.50 ± 1.82 ^ab^	52.98 ± 9.53 ^ab^	35.60 ± 0.49 ^b^	10.50 ± 1.06 ^b^

x: RI, the retention index on capillaries DB-WAX. ND: not detected. y: the relative concentration stated as the mean ± SD and the unit is ppt (part per trillion, ng/g).^a,b,c,d^: Different letters represent significant data, while the same letter represents weak significance.

**Table 9 molecules-27-09047-t009:** The weighted score of each flavor type.

Odor Type	Weight
Milky	0.3
Muttony	0.2
Animal fat	0.2
Sweet	0.3
Fruit	0.2
Burnt	0.1
Spoiled	−0.2
Sour	−0.1

## Data Availability

The authors will make the raw data supporting the conclusions of this manuscript available to any qualified researcher.

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
