# Peer review of "Optimization and Preparation of Tallow with a Strong Aroma by Mild Oxidation"

_molecules, 2022, doi:10.3390/molecules27249047_

Round 1

Reviewer 1 Report

The presented topic is original and interesting. The authors have made an effort to conduct a reliable qualitative and quantitative analysis. Nevertheless, there is a lack of information and I have suggestions to improve the manuscript.

1.     Add some essential data of qualitative and quantitative analysis from the odors study to the abstract.

2.     Lines 111-112: add space before min. Check the whole manuscript

3.     Add references to Food Safety Standards. 

4.     Materials and Methods: add a paragraph about Chemicals where you put a name, CAS, purity, and supplier of all reagents that you used.

5.     In which solvent was prepared 1-hexanol?

6.     Description of the SPME method should be written in the past tense, for example, The carrier gas was….Check the whole manuscript

7.     Add the Brand and model of GC-MS

8.     Solute delay or solvent delay?

9.     Why do you use solvent delay in spme method? Don’t you smell the most volatile compounds during the first 4 min of the chromatogram?

10.  Which equipment did you use for smelling? Sniffing port? From which company?

11.  2.8. Statistical analysis (capital letter)

12.   Add a description of how quantitative analysis was performed

13.  Name well the analytical methods or SPME-GC-MS or SPME-GC-O-MS

14.  Which program did you use for PCA? Add to the statistical part

15.  Figure 4 improve the names of compounds as they are overlaid

16.  Figure 8 add errors to all data

Author Response

Dear reviewer

The manuscript is revised according to your comments, please have a check. If more are needed, please let me know, and we are sure to do our best to meet your requirement.

Sincerely Yours

Huanlu Song

Author Response

Dear Reviewer

The manuscript is revised according to your comments, please have a check. If more are needed, please let me know.

Sincerely Yours

Huanlu Song

Round 2

Reviewer 1 Report

The manuscript has been significantly improved. I don´t have any more suggestions.

Reviewer 2 Report

Thank you for your response. I have no comments anymore.